# Key determinants of global land-use projections

Elke Stehfest[1], Willem-Jan van Zeist[1], Hugo Valin [2], Petr Havlik[2], Alexander Popp[3], Page Kyle [4], Andrzej Tabeau[5], Daniel Mason-D'Croz [6,7], Tomoko Hasegawa [2,8,9], Benjamin L. Bodirsky [3], Katherine Calvin [4], Jonathan C. Doelman[1], Shinichiro Fujimori [2,8,10], Florian Humpenöder[3], Hermann Lotze-Campen [3,11], Hans van Meijl[5] & Keith Wiebe [6]

Land use is at the core of various sustainable development goals. Long-term climate foresight studies have structured their recent analyses around five socio-economic pathways (SSPs), with consistent storylines of future macroeconomic and societal developments; however, model quantification of these scenarios shows substantial heterogeneity in land-use projections. Here we build on a recently developed sensitivity approach to identify how future land use depends on six distinct socio-economic drivers (population, wealth, consumption preferences, agricultural productivity, land-use regulation, and trade) and their interactions. Spread across models arises mostly from diverging sensitivities to long-term drivers and from various representations of land-use regulation and trade, calling for reconciliation efforts and more empirical research. Most influential determinants for future cropland and pasture extent are population and agricultural efficiency. Furthermore, land-use regulation and consumption changes can play a key role in reducing both land use and food-security risks, and need to be central elements in sustainable development strategies.

[1] PBL Netherlands Environmental Assessment Agency, Postbus 30314, The Hague, 2500 GH, The Netherlands. [2] International Institute for Applied System Analysis (IIASA), Schlossplatz 1, Laxenburg 2361, Austria. [3] Potsdam Institute for Climate Impact Research (PIK), P.O. Box 60 12 03, Potsdam 14412, Germany. [4] Joint Global Change Research Institute, Pacific Northwest National Laboratory, 5825 University Research Court, Suite 3500, College Park, 20740 Maryland, USA. [5] Wageningen Economic Research, Wageningen University and Research, P.O. Box 29703, The Hague, 2502 LS, The Netherlands. [6] International Food Policy Research Institute (IFPRI), 1201 Eye St., N, Washington DC 20005-3915, USA. [7] Commonwealth Scientific and Industrial Research Organisation (CSIRO), St. Lucia, Queensland, Australia. [8] Center for Social and Environmental Systems Research, National Institute for Environmental Studies (NIES), Tsukuba, Japan. [9] Department of Civil and Environmental Engineering, Ritsumeikan University, 1-1-1, Nojihigashi, Kusatsu, Shiga 525-8577, Japan. [10] Department of Environmental Engineering, Kyoto University, Kyoto, Japan. [11] Humboldt-Universität zu Berlin, Berlin, Germany. Correspondence and requests for materials should be addressed to E.S. (email: Elke.Stehfest@pbl.nl)

Scenarios of land use and land cover play an important role in exploring future developments and policy options for climate change, biodiversity, food security, ecosystem services, and sustainable development. The exploration of future global land use in international assessments started in the 1990s (SRES)[1] and gained increasing importance in environmental outlooks and assessments[2,3]. During the last decade, the number of studies and models on global land-use projections has increased tremendously, reflecting concerns about land scarcity, climate change impacts or bioenergy threatening food security[4–7], loss of natural areas and biodiversity, and sustainable development in general[8,9]. However, despite the central role of land use in future environmental change, the modeling and systematic model comparison of global land-use projections is still in its infancy[10,11] and the uncertainty in results is large[12–15]. Land use results from multiple interactions between regionally specific demand and supply systems, numerous feedback processes, and smaller-scale factors such as land-use regulation and land ownership, and therefore projecting land-use change is highly complex. As an example, although agricultural production has increased by about 60% during the last 40 years, global cropland area has increased by only 5% as a result of increased agricultural productivity (intensification)[16]. Therefore, the intricate interplay between demand and production, and agricultural intensification is a core determinant of future land use. To account for the range of possible developments in these drivers, long-term projections are often dealt with following a scenario approach, where fundamental uncertainties in socio-economic factors are explicitly varied along so-called storylines to explore contrasting futures. However, the range of model outcomes for such scenarios tends to be large and results may depend equally on model characteristics as on storylines and assumptions[15]. Given the importance of land-use projections for informing policy makers in the areas of climate change, food security, and biodiversity protection, it is important to investigate the spread across storylines and model results, to better understand the possible evolution of land use and the food system.

Here we use the scenario framework of the Shared Socio-economic Pathways (SSPs)[17] and their implementation by integrated assessment and agricultural models[14], to explore how long-term drivers determine projections of land use and food availability. We follow a sensitivity methodology recently applied to projections of $CO_2$ emissions of the SSP scenarios[18] to assess the contribution of each driver to the scenario outcomes. This allows us to explain model spread and results and, more importantly, to identify the key determinants of future land use, their relative importance, and interactions. Although single model studies have identified factors that determine future land use and support a transition to a sustainable land and food system, this study allows a systematic comparison across factors and models, and identifies the interaction between factors. Furthermore, the discussion highlights gaps and shortcomings in the current modeling of global land use to guide future research priorities.

The SSP scenario framework consists of five contrasting storylines[19], with SSP2 representing a baseline development with continuation of current trends, a sustainability scenario (SSP1), a regional rivalry scenario (SSP3), a fragmentation scenario (SSP4), and a fossil fuel scenario (SSP5). Here we focus on the socio-economic developments of SSP1, SSP2, and SSP3 as their land-use implications were initially explored in most detail and by the largest number of models[12,14]. Results related to SSP4 and SSP5 are however also available. The SSP scenarios play a crucial role in the ongoing assessment and reports by The Intergovernmental Panel on Climate Change (IPCC)[20], in the agricultural modeling community AGMIP[6,21,22] and in the Inter-Sectoral Impact Model Intercomparison Project ISIMIP[23], and also in assessments of biodiversity within IPBES[24] and probably a range of future assessments of sustainable development. The SSP narratives represent contrasting global developments with respect to population growth, economic development, technological change, consumption preferences, environmental protection, and international cooperation (see Methods and Supplementary Information). Although projections for population and GDP were harmonized in quantitative terms within the SSP process[25,26], all other storyline elements were translated into scenario drivers by each team separately, because differences in model structure hampered quantitative harmonization.

Our analysis covers all five models that participated in the initial quantification of land use in the SSPs[14], namely AIM[27], GCAM[28], GLOBIOM[29], IMAGE-MAGNET[30,31], and MAgPIE[32], plus an additional model, IMPACT[33], frequently used in agricultural assessments[3,34].

To determine the sensitivity of scenario outcomes to the scenario drivers, we rely on a sensitivity analysis protocol, which allows to identify the interaction between drivers[35] and which was previously applied to the analysis of $CO_2$ emission trajectories as well[18]. We distinguish six groups of scenario drivers: population growth (POP), economic growth in gross domestic product per capita (GDPpc), land-use regulation (LUR), agricultural productivity growth (PRD), consumption preferences (CON), and trade development (TRD). Using a matrix of scenario experiments, the deviation of other SSPs from the SSP2 baseline is attributed to these six driver groups, showing their individual, final, and interaction effect (Supplementary Fig. 1). The individual effect describes how scenario results change if a single factor of SSP2 baseline is replaced by the one used in SSP1 (or SSP3) and the final effect shows how results in the full SSP1 (or SSP3) scenario change if this single factor is changed back to the SSP2 default setting; the interaction effect is computed as the difference between individual and final effect, and shows how other factors interact with this single factor, i.e., reduce the individual effect to the final effect (see Methods). In order to analyze how sensitivity results depend on the quantity of a driver, we derive from the experiments the size of the drivers and relate them to the change in outputs.

We find that spread-across models arises from diverging sensitivities to long-term drivers and from various representations of land-use regulation and trade, showing the need for reconciliation and more empirical research. We identify population and agricultural efficiency as most influential drivers and show that land-use regulation and consumption changes can play a key role in reducing both land-use and food-security risks. The analysis of interactions between factors reveals that efficiencies, land-use regulation, and consumption change are most effective when ccombined, and all three need to be central elements in sustainable development strategies.

## Results

**Spread in model results**. The spread in model results for land use and the agricultural system is large but some common patterns emerge (Fig. 1 and Supplementary Fig. 2). In all models, cropland area in 2050 is lower in SSP1 (−13% on average) and higher in SSP3 (+8% on average), compared with SSP2 (Fig. 1). This is consistent with the storylines that describe more sustainability and regard for environmental boundaries in SSP1, and stronger environmental degradation in SSP3. Similarly, out of the five models reporting pasture area, four find a decrease in SSP1 (−8% on average) and four find an increase in SSP3 (+2% on average), compared with SSP2 until 2050. Due to lower economic growth in SSP3, per-capita food demand decreases in SSP3 for five of the six models, by up to 10% compared with SSP2 until 2050. The

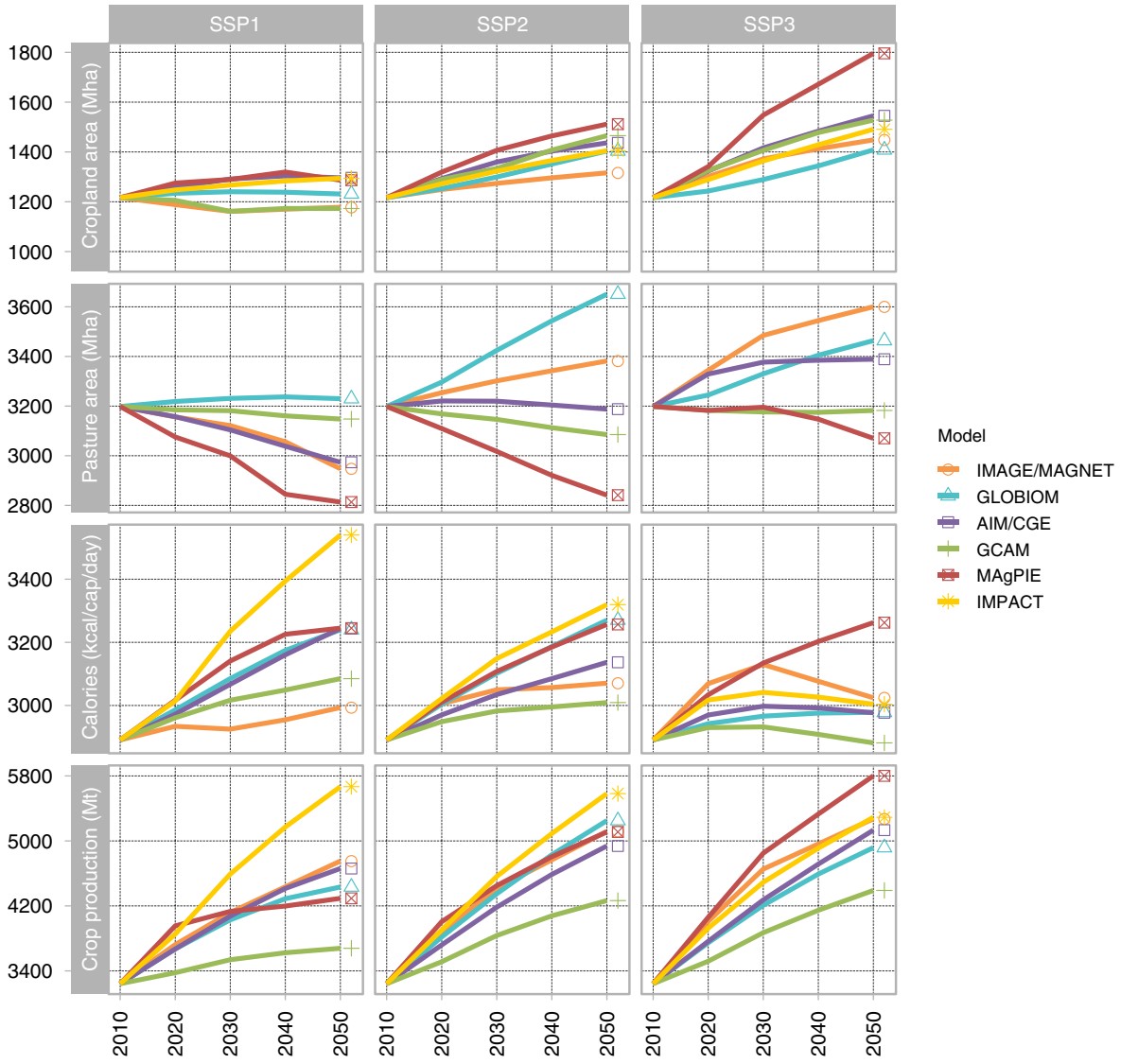

**Fig. 1** Key characteristics of global agriculture and land use in SSP1, SSP2 and SSP3. Model results were standardized to a common value in 2010. The order of models in the legend follows the publication of land use in the SSPs, with SSP1, SSP2, SSP3, SSP4 and SSP5 being represented by IMAGE, GLOBIOM, AIM, GCAM and MAgPIE, respectively[14]

picture is more heterogeneous for SSP1 with three models reporting an increase in 2050 and three a decrease compared with SSP2. As a combined effect of population, per-capita food demand, and other scenario-specific drivers, crop production (for food and feed) decreases for five models in SSP1 (−10% on average) and increases in SSP3 for four models (+2% on average). These projections are consistent with those observed in Popp et al.[14].

**Disentangling the drivers**. The results of our sensitivity analysis (individual effect, final effect, and interaction effect by driver) are presented to understand changes in cropland (Fig. 2) and food consumption (Fig. 3), for SSP1 and SSP3. Pasture and food production are also provided in the Supplementary Information (Supplementary Figs. 3 and 4). Supplementary results for trade, prices, livestock consumption, and productivities are also provided in the Supplementary Fig. 5 but not covered by our discussion.

**Global cropland**. The sensitivity analysis on cropland shows that the most important drivers are population (POP), consumption

preferences (CON), and agricultural productivity growth (PRD) (Fig. 2). Economic growth (GDPpc) has a smaller effect and shows an opposite effect to the other scenario factors in SSP1 and SSP3 on crop area (e.g., lower population in SSP1 results in lower cropland area, yet GDPpc is higher in SSP1, which has the opposite effect). The impact of the various factors depend on the driver sizes (across models and across scenarios), as analyzed further below. The change in cropland area as a result of agricultural productivity growth (PRD) varies across models, as they differ in how agricultural technological improvement affects either area use or consumption volumes. Land-use regulation (LUR) can have a substantial contribution to changes in cropland area, as seen in IMAGE/MAGNET and IMPACT. The apparently small sensitivity of scenario outcomes to LUR in other models is related to little or no changes in this driver compared with SSP2, as shown by the analysis of drivers size (Supplementary Fig. 6), and models vary largely in their implementation of land-use regulation (Supplementary Data 1). Through availability of area for food production, LUR also affects agricultural production and food demand (Fig. 3), and up to 60% of the theoretical production change through limited crop area is compensated by increased

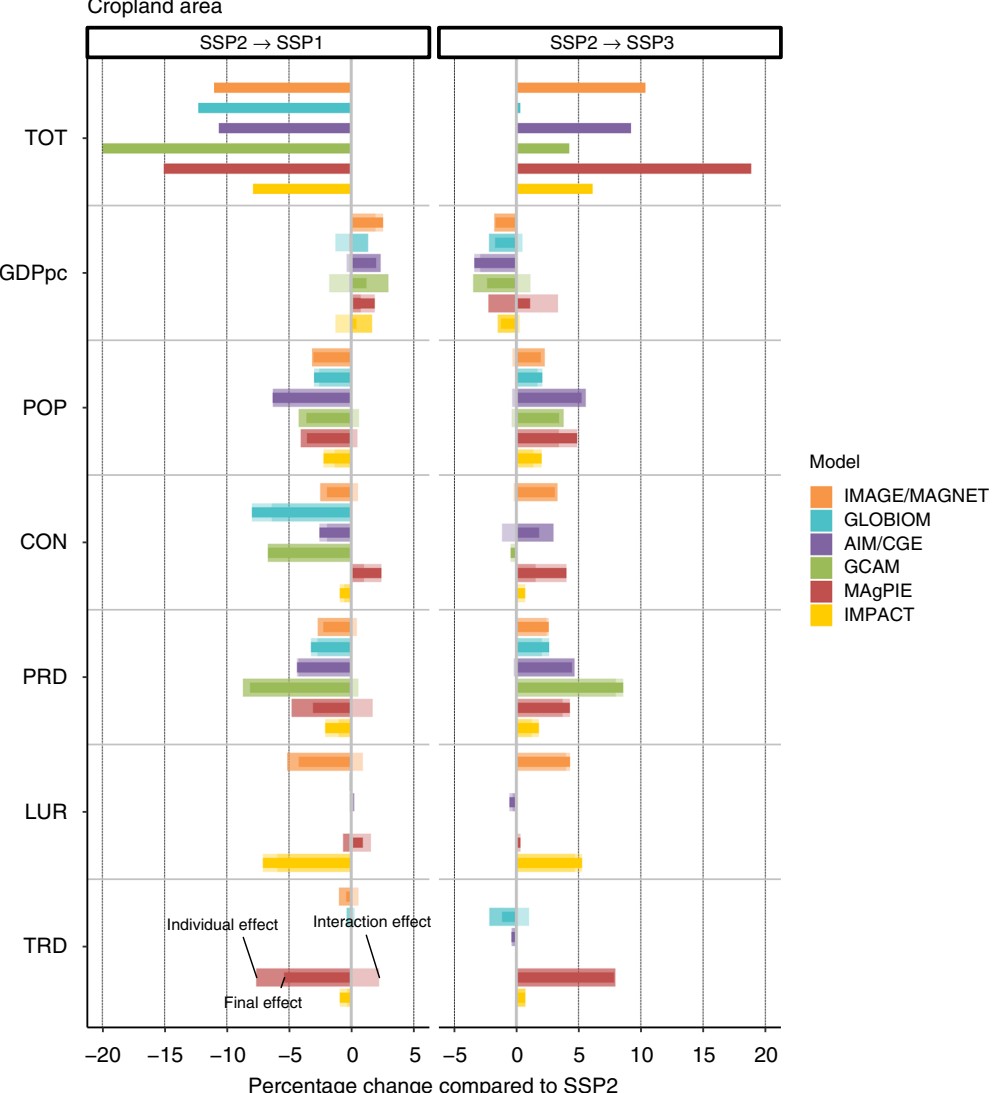

**Fig. 2** Sensitivity of cropland area to groups of drivers. Changes refer to change in cropland area in 2050 when moving from SSP2 to either SSP1 (SSP2- > SSP1) or SSP3 (SSP2- > SSP3). Individual effects (wide, light-colored bars), final effects (thin, dark-colored bars), and interaction effects (wide, very-light-colored bars) are shown for each of the six driver groups (CON: consumption preference, GDPpc: economic development, LUR: land-use regulation, POP: population, PRD: agricultural productivity growth, TRD: trade development), together with the total difference between SSP1 and SSP3, respectively, and SSP2 (TOT). The factor CON in MAgPIE also includes restrictions to irrigation (see Supplementary Information, Driver implementation)

agricultural efficiencies. The spread in model results is mostly explained by diverging effects of CON and PRD on cropland area in the various models, and by one model (MAgPIE) expecting large differences between scenarios due to trade development (TRD).

**Pasture land**. Pasture area is most sensitive to PRD, CON, and LUR, although the strength of the effect across the drivers is very heterogeneous across models (Supplementary Fig. 3). One model (GLOBIOM) projects a decrease in pasture in SSP3 compared with SSP2, mainly due to a reduction in total production, which is caused by lower GDPpc, lower PRD, and TRD effects. In other models, the tendency toward increasing pasture through lower productivity and higher consumption and production dominates.

**Food demand**. Population (POP) and economic development (GDPpc) are the most important determinants for food demand in SSP1 (SSP3), with lower (higher) population and higher

(lower) economic growth leading to more (less) per-capita demand. Population increases pushes down per-capita consumption due to higher food prices, as high population growth increases pressure on land and production. Thus, for per-capita consumption, trends of POP and GDPpc work in the same direction, while they have a compensating effect in all other variables where the size of the population plays a role (e.g., total production and total land use). Stricter regulation of land use (LUR) appears in several models as an effective measure to limit agricultural area (see above) but leads to lower food consumption in SSP1, thus increasing the risks to food security. The interaction effect, e.g., in IMAGE/MAGNET, however, shows that the additional scenario drivers of SSP1 (consumption change (CON) and higher agricultural productivity (PRD)) reduce this adverse effect of LUR, while maintaining the beneficial effect on cropland extent. The storylines explicitly describe lower (higher) environmental impact of food consumption in SSP1 (SSP3), reflected in lower (higher) meat consumption and food waste, respectively. Caloric consumption per capita in SSP1 is very sensitive to these

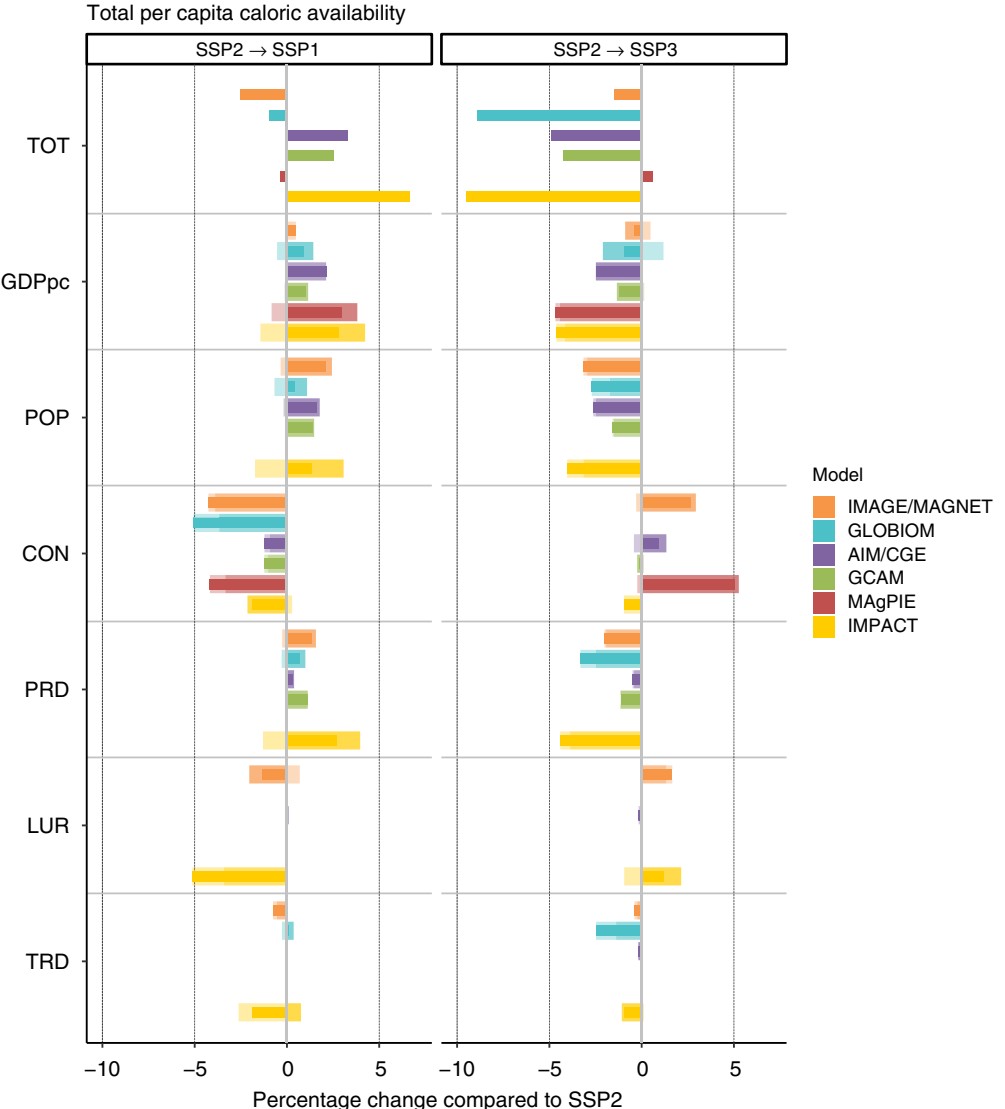

**Fig. 3** Sensitivity of consumption per capita to groups of drivers. Changes refer to consumption change in 2050 when moving from SSP2 to either SSP1 (SSP2- > SSP1) or SSP3 (SSP2- > SSP3). Individual effects (wide, light-colored bars), final effects (thin, dark-colored bars), and interaction effects (wide, very-light-colored bars) are shown for each of the six driver groups (CON: consumption preference, GDPpc: economic development, LUR: land-use regulation, POP: population, PRD: agricultural productivity growth, TRD: trade development), together with the total difference between SSP1 and SSP3, respectively, and SSP2 (TOT)

assumptions on consumption preferences (CON), but less so in SSP3, as this element of the storyline was implemented less stringently in SSP3 (Supplementary Fig. 6). Differences in trade development (open, globalized trade in SSP1 and restricted, regionalized trade in SSP3) hardly affect caloric consumption per capita (Fig. 3). Faster progress in agricultural productivity (PRD) improves caloric consumption in SSP1, whereas slower progress decreases it in SSP3, in the models with endogenous food demand response to prices. Evaluating caloric consumption separately for crops and livestock products (Supplementary Fig. 2) reveals that models show a rather homogenous picture in livestock consumption as an effect of consumer preference changes, at the global level, but diverge on per-capita consumption of crops as a result of diverse assumptions on waste reduction and products substituting animal-sourced food.

**Crop production**. Change in crop production in SSP3 compared with SSP2 is largely explained by population (POP) and economic development (GDPpc) (Supplementary Fig. 4). Where consumption preference (CON) or trade development (TRD) has a large effect, the decrease in SSP1 can be more than 10%, mostly due to reduced consumption of livestock commodities or trade shifts to more efficient production systems, and consequently reduced feed demand. Faster progress in agricultural productivity (PRD) triggers higher demand and production of crops, mediated via reduced prices. The interaction effect between groups of drivers is large in some cases. For example, a strong individual effect of GDPpc on total production is much less important when consumption preferences change toward less livestock products (GLOBIOM). Although agricultural production is projected to decrease in SSP1 compared with SSP2 in almost all models, the direction of change is less clear for SSP3, as the increasing effect of population and dietary preferences is counterbalanced by the reducing effects of GDPpc and agricultural productivity. In cases where a decrease of crop production is projected for SSP3, a large sensitivity to agricultural productivity is present (GLOBIOM and IMPACT).

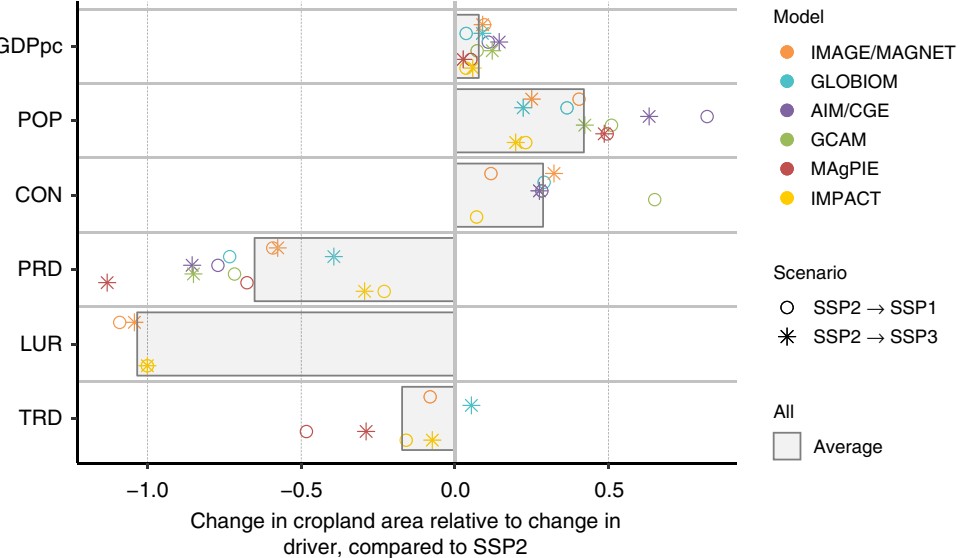

**Fig. 4** Elasticity of cropland area change to a change in scenario driver. The elasticity indicates how a relative change in the driver, e.g., population, translates to a relative change in output variables, in this case cropland area. For example, we see that if PRD would increase in GCAM by 10%, cropland area would be decreased by about 8% (elasticity of −0.8), whereas the same change of 10% in PRD in IMPACT would lead to a reduction of about only 2% (elasticity of −0.2). Data points per model and scenario show average across final and individual effects. Drivers' abbreviations: CON: consumption preference, GDPpc: economic development, LUR: land-use change regulation, POP: population, PRD: agricultural productivity growth, TRD: trade development. We excluded data points of which the underlying changes were very small (<0.5%)

**Driver size**. SSP1 and SSP3 do not differ equally from SSP2, not in terms of total results and not in terms of sensitivity to drivers (Supplementary Figs. 5 and 7). This can be due to nonlinear model dynamics, but is also caused by driver sizes for SSP1 and SSP3 not differing equally from SSP2 (Supplementary Fig. 6). Driver sizes also differ across models, as they were quantified independently and as they are influenced by modelers' interpretation and current knowledge about possible future developments, e.g., in agricultural technology (Supplementary Fig. 6). To analyze the effect of driver sizes on the sensitivity analysis results, we relate for each sensitivity test the change in output to change in driver magnitude, so that a general elasticity of output variables to driver inputs can be assessed across models and scenarios (Fig. 4). An increase in population and livestock consumption (CON) leads to an increase in cropland area and an increase of crop productivity leads to a reduction of cropland area with a relative effectiveness of more than 50% on average. Interpretation of the effectiveness of land-use regulation (LUR) is difficult due to problems in measuring the size of the driver and needs to be improved in future assessments. Although analysis of these elasticities reveal additional insights into model behavior and how sensitivities to a driver propagate through the individual models, it however does not inform on possible future variation, e.g., GDPpc seems to have a very limited effect compared with population (POP), but the possible variation over the next 50 years in GDPpc is much larger than for population, as also reflected in the SSP projections.

## Discussion

This analysis has shown that global land use could evolve in diverging directions in the future, either following the historical expansion of cropland and pasture, or moving toward a reversing trend in agricultural land use. These future developments will have multiple implications for greenhouse gas emission, biodiversity, and potential for carbon uptake and bioenergy production. Decomposing drivers of change, we find that the most influential ones would be changes in agricultural productivity,

consumption preferences, and population development. Land-use regulation also has a large potential to affect future land use, but is currently underrepresented in models and should receive more attention in future assessments. The possibly beneficial effects of increased trade on global land use are uncertain and partly contradict earlier findings[36], and also require further research. Some other drivers included in the SSPs storylines may also deserve scrutiny, but are not yet quantified by global land-use models, due to technical difficulties in quantifying their impact (urban migration, inequalities, farming structure, and governance). Beyond the global dynamics addressed here, regionally diverging trends may lead to smaller effects (e.g., for dietary change due to leakage[37]), or to different results for the controversial land-saving of higher crop yields[38]. Furthermore, consistent with the SSP scenario logic[17], we considered here only socio-economic drivers and did not include climate impact and mitigation elements, such as yield impacts from change in temperature and precipitation, $CO_2$ prices, or afforestation deployment for carbon sequestration. Analysis elsewhere has shown the potentially large area demand of land-based mitigation[14,39], higher intensification of croplands, and possible interaction with the food and agricultural system[6].

Differences across models can be explained by two main causes. First, modelers have differing quantitative interpretation of storylines. Each modeling team follows their own assumptions in terms of future productivity change, extent of diet shifts, or area protection, depending on their own assessment of past developments and possible future evolution. In some cases, drivers are not or poorly represented, such as land-use regulation, although it has a large potential to restrict future agricultural land expansion[40]. Currently, global models tend to only exclude certain areas from agricultural conversion, mostly focusing on official protected areas and their assumed expansion according to international targets, but ignore specific national and sub-national regulations on land-use zoning or deforestation rates (e.g., sugar cane zoning and forest code land reserve quotas in Brazil) and, as also shown here, land-use regulation is necessary to assure the land-saving effect of agricultural intensification[38]. Although the

effectiveness of such approaches can be poor[41], these regulations and land-tenure aspects are relevant to short- and medium-term conversion quantities and locations. Therefore, future model development should include more explicitly land-use regulation and its facets at various spatial scales, and also its interaction with agricultural productivity and food security. Changes in international trade is another example of parsimonious implementation, with some models considering solely adjustments to current tariffs, whereas some others assume more fundamental restrictions and amplification of global trade in a long-term perspective.

The second cause for differences in model results comes from parameterization choices and input data selection. These would gain from being further compared, harmonized, and improved through model intercomparison exercises[12]. However, differences in model design also play a role and reflect the various areas of focus between the tools (geochemical cycles, agricultural markets, and economy-wide analysis) and the representation of system behavior[12]. For instance, drivers behind land productivity change (prices, research investments, technology adoption, etc.) can greatly vary depending on whether the model is primarily dealing with agricultural development or deforestation. Demand feedback[42] can be omitted in models focusing on production optimization, but in this analysis we see that the spread in results also arises from diverging sensitivities in demand feedbacks, even missing in one of the models. To cope with diversity of model views, use of ensemble analyses is recommended rather than single model assessments and these now became mainstream in climate policy[39], food[6] and biodiversity[43], and in the SSP analysis of future land use[14]. This study highlights the different sensitivity of the model responses to scenario drivers and could be used to improve the model design in the future, develop recommendation and best practices, and guide users of the model results when comparing individual model results against the whole ensemble.

Although global land-use models have the ambition to include the relevant drivers and processes to project future large-scale land-use dynamics, the identification of key determinants of future land use needs to account for aspects currently poorly addressed by the models. Local land-management regulations, as discussed above for protected areas, are poorly represented in current models, although they can play important roles at fine scale. Land tenure, processes of intensification and technology transfer, and integration of crop cultivation and livestock with forestry and other land uses (urbanization, recreational areas, and mining) are other underrepresented factors relevant for future land projections. To identify the gaps and shortcomings in global-scale models, more rigorous model validation and evaluation on smaller scales are necessary.

Some of the factors that we identified as key elements for future land use have been analyzed earlier. Single model studies have highlighted the importance of dietary transitions[44–46], increased crop yields[47], and livestock efficiencies[37,48,49], or have shown the combined effects of demand side, production, and land-use regulation[8,9]. Beyond these earlier studies, this systematic multi-model analysis allows to compare the relative effect across the drivers and across the models. Furthermore, this study adds an analysis of interactions between factors: an increase in agricultural efficiency can reduce demand for cropland but can also trigger higher demand. Yield increase is therefore shown to be more effective in reducing land use when combined with land-use regulation and consumption changes. Similarly, land-use regulation can decrease expansion into cropland but may form a risk to food security, if not bundled with increased productivity and consumption shifts.

Consumption changes, better land management, and increased agricultural productivity therefore appear robust and effective measures to reduce the demand for agricultural land, in particular

when bundled together. As land-use development is closely tied to challenges related to biodiversity, food security, or climate, the targeting of these measures appear as a key component of the long-term sustainability agenda.

## Methods

**The SSP narratives**. To explore possible futures of global environmental change and climate policy, the scientific community has developed the so-called SSPs. The narratives behind these scenarios have been described in qualitative terms[19] and were recently quantified in integrated assessment models[17]. Designed to cover the challenges space for mitigation and adaptation, they describe five consistent pathways along which the world could develop and address all relevant aspects from population, economic development, education and health, and environmental awareness to international cooperation (Supplementary Table 1). In this study, we focus on SSP1, SSP2, and SSP3 (SSP4 and SSP5 shown in Supplementary Fig. 8), which describe a world of low, medium, and high challenges to both mitigation and adaptation, respectively. The sustainability scenario SSP1 is characterized by low population growth, fast and inclusive economic growth, respect for environmental boundaries, low environmental impact of consumption, fast technological development, and global cooperation. SSP2 shows continuation of current trends in all aspects, whereas the regional rivalry scenario SSP3 is dominated by high population growth, slow economic development, larger environmental impact of consumption, slow technological progress, and regional fragmentation and trade barriers.

The development of land use in these scenarios is determined by trends in the factors described below; as model results change when the input of these factors changes according to the SSP-specific storyline, these were varied in this sensitivity analysis.

POP describes the population development per region over the scenario period and was quantified for the SSPs by ref. [25]. SSP1 has lower population growth in most regions, whereas population growth in SSP3 is high in developing regions and low in developed regions.

GDPpc describes the regional income per year and per capita, over the scenario period, and as derived for the SSPs by a long-term growth model[26] based on the population projections described above. Per-capita income grows faster in SSP1 than in SSP2, also due to faster demographic transition, whereas growth is slower in SSP3.

CON describes a range of consumer preferences with respect to environmental impact of food consumption. In SSP1, orientation toward lower resource intensity leads to reduced demand for livestock products and to a reduction of waste in the food system, whereas in SSP3 an opposite development dominates.

PRD describes changes in the productivity of agricultural systems over the scenario period. In SSP1, faster technological development and respect for environmental boundaries leads to increased efficiency in agricultural systems, including higher crop yields and more efficient livestock systems, whereas in SSP3 technological process in agricultural systems is rather slow.

LUR describes land-use regulation as part of the SSP scenarios. In SSP1, land-use regulation is described as rather strong and weak in SSP3, which is reflected both in the area designated for nature conservation, but also in the ease of conversion from natural to agricultural land.

TRD describes how international trade develops over the scenario period. SSP1 is characterized by globalized trade and low trade barriers, but also by increased preference for regional production in food consumption. Contrary, SSP3 is characterized by higher trade barriers and regional fragmentation.

The precise implementation of these six sets of assumptions across the three SSP scenarios varies in the six models (see further below). These sets are referred as sensitivity factors, as each represents a collection of related scenario features relevant to the sensitivity. Bioenergy demand is not included in this analysis and does not vary across the SSP scenarios of one specific model.

**Models used**. In this study we included all five Integrated Assessments Models (IAMs) involved in the recent SSP quantification[17] (either the full IAM or its land component). These models are state-of-the-art global climate–energy–economy models including a land-use component. Their quantification of land-use futures[14] is now being used in various assessments, e.g., within IPCC and IPBES. One additional model, the IMPACT model[33], which is frequently applied in global assessments of land use and food security[3,5,6,34], also contributed results to the sensitivity experiments. Here we give a brief description of the models:

AIM[27] is a large-scale computer simulation model and represents the energy and land systems in one computable general equilibrium (CGE) structure. Supply, demand, investment, and trade are described by individual behavioral functions that respond to changes in the prices of production factors and commodities, as well as changes in technology and preference parameters. Production functions are formulated as multi-nested constant elasticity substitution functions. Household demand is formulated as a linear expenditure system function. A single international trade market is assumed for each traded commodity and substitution between domestic and imported commodities is based on the Armington assumption. Allocation of land by sector is formulated as a multi-nominal logit

function[27]. The model contains 17 regions and 42 sectors, including 10 agricultural ones.

GCAM[28] is an integrated assessment model comprising modules for the economy, the energy system, the agriculture and land-use system, and for climate. The agriculture and land-use component[50] determines supply, demand, and prices for crops, livestock, forest products, and bioenergy in an iterative procedure to achieve market equilibrium, in interaction with agents and prices of the economy and energy system. Supply and land use is resolved for 283 world regions, and within each of the these model regions land is allocated economically based on profit maximization with an assumption of nonlinear distributions of profits for each competing use. Demand for agricultural commodities is resolved for 32 world regions, based on population, income, and price levels. GCAM allows for global trade in crops, forestry, and bioenergy. GCAM distinguishes 12 crop and forestry categories, 6 animal categories, and bioenergy, in 32 or 283 world regions for demand and production, respectively.

GLOBIOM[48] is the land component of the MESSAGE-GLOBIOM integrated assessment model. It is a global partial equilibrium model covering agriculture, forestry, and bioenergy as the three main competing land-use sectors. For its production processes, the model uses detailed grid-cell information on biophysical and technical cost information. Final demand is computed for households, industry, and services through own-price elasticities demand curves specific to each product. GLOBIOM has a spatial equilibrium modeling approach and represents bilateral trade based on cost competitiveness. The model has been used for integrated assessment of climate change impact and mitigation policies in land-based sectors, agricultural and timber markets foresight, analysis of food security in the long term, and its trade-offs with environmental indicators. The global version of GLOBIOM covers 30 regions and 25 agricultural commodities (18 crops and 7 livestock commodities), as well as bioenergy products.

IMAGE-MAGNET identifies land-use scenarios from the integrated assessment model IMAGE[30] using the CGE model MAGNET[31] to represent the agricultural economy. MAGNET is a multi-regional, multi-sectoral, applied general equilibrium model based on neo-classical microeconomic theory, and it is an extension of the standard GTAP model[51]. The agricultural sector is represented in high detail compared with standard CGE models. Land supply, suitability, and potential agricultural yields are evaluated on a grid-scale in IMAGE and provided to MAGNET and used to model land as a production factor described by a land supply curve[52]. Other production factors are labor, capital, and chemical inputs. Household demand depends on population, income, and income and price elasticities, whereby price elasticities change with increasing incomes, and are also influenced by consumer preferences. Trade is modeled bilaterally using the Armington assumption[53]. IMAGE and MAGNET use 26 world regions and 10 agricultural commodities.

IMPACT is a partial equilibrium model used to analyze long-term challenges and opportunities for food, agriculture, and natural resources at global and regional scales[33]. It links information from climate models (Earth System Models), crop simulation models (e.g., Decision Support System for Agrotechnology Transfer), and water models to a core global, partial equilibrium, multimarket model focused on the agriculture sector. Future production is influenced by price-driven changes in productivity and land use, as well as by external trends of productivity and land dynamics. Demand for agricultural commodities is a function of the price of the commodity and the prices of other competing commodities, per-capita income, total population, and consumer preferences, which are represented by price and income elasticities of demand, based originally on estimates by the United States Department of Agriculture, and adjusted to represent various future dietary scenarios and to satisfy Engel's and Bennett's Laws. Trade is represented as pooled trade and IMPACT distinguishes 62 primary and processed commodities, and 159 countries (for demand) and 320 Food Producing Regions.

MAgPIE 3.0[32,54] is the land-use component of the REMIND-MAgPIE integrated assessment modeling framework[55]. MAgPIE optimizes global agricultural production costs under consideration of biophysical and socio-economic constraints, including land-use-related policies. MAgPIE estimates crop production patterns and pasture areas on spatial resolution of clustered 0.5° grid cells, using crop-physiological and hydrological data from LPJmL[56]. Food demand for plant and livestock calories is estimated using an econometric regression model[57]. Functional forms of the demand system can be adapted to the storylines. Demand systems account for income but not for price elasticity. Feed demand for livestock production is considering regional livestock-specific feed baskets that are dynamic when livestock productivity increases over time[49]. International trade is estimated based on cost-efficiency, but is constrained by a scenario-specific degree by historical trade patterns[58]. Crop yields can be endogenously increased by investments into research and technology[59]. MAgPIE has 10 world regions and differentiates 17 crop types, 5 livestock products, 2 bioenergy crops, pasture and crop residues, oilcakes, and molasses.

The model-specific implementation of the SSP drivers is described in the next section. Additional model documentation can also be found online http://themasites.pbl.nl/ models/advance/index.php and in the individual model documentations.

### Implementation of SSP drivers in the models

The implementation of the SSP storylines into the models AIM[27], GCAM[28], GLOBIOM[48], IMAGE-MAGNET[30,31],

and MAgPIE[32,54] has been described in detail in Popp et al.[14]. Here we present an overview of this implementation across models including IMPACT[33] and also address specific issues of implementation in the various models (Supplementary Data 1). The quantification of drivers is described further below.

GDP per capita (GDPpc) is implemented in strongly different ways across the models. Partial equilibrium models (GCAM, GLOBIOM, IMPACT, and MAgPIE) have GDPpc only influencing the consumer income and therefore the demand part of the model. Contrary, the general equilibrium models (AIM and MAGNET) have GDP influence not only income but also technological change and thus also the overall structure of the economy and the supply side. Therefore, these models estimated shifters of technology change (capital and labor productivity) according to the exogenous GDP (i.e., GDPpc × population) and compute technological change endogenously. This procedure is applied in the scenario of GDPpc (_GDPpc) and POP (_POP), as both types of scenarios change total GDP and technological change.

Population (POP) according to the SSP-specific projections[25] is implemented straightforward in the partial equilibrium models GCAM, GLOBIOM, and IMPACT, and affects the entire demand system and interaction with price effects and production and trade. In MAgPIE, demand per capita is not effected by population changes due to absent price feedbacks in the demand system[57]. In the CGE models AIM and MAGNET, population size also changes the total GDP, via scenario-specific GDPpc, and thus the technological change calibrated to meet this total GDP; in these models, population affects the agricultural system both via the demand side and also the technology side.

Consumption preferences (CON) are described in the SSP storylines as differences in consumer behavior toward environmental impact of food consumption via food waste and share of animal products in the diet. The implementation of food waste is straightforward in all models, although with varying fractions of waste reduction. In addition, the consumption of animal products was reduced in all models in SSP1 and increased for SSP3 in some models, mostly via preference factors. Some models also included health guidelines in defining their diets for SSP1. In addition to different food-demand assumptions, the MAgPIE implementation of CON includes assumptions on soil nitrogen uptake efficiency and environmental flow protection. In CON for SSP1, the latter limits the available water for irrigation and leads to a substantial decrease in crop yields and larger crop area. For this reason, MAgPIE results were excluded from Fig. 4.

Agricultural productivity (PRD) is characterized by fast technological change and respect for environmental boundaries for SSP1 and slower technological change in agricultural systems in SSP3. For crop systems, models mostly adjusted the progress in (potential) crop yields to reflect the storylines or linked this progress to scenario-specific GDP growth. For livestock systems, whose efficiencies are much more difficult to compare, they followed a similar approach or adjusted the speed in which systems can catch up to the most efficient ones.

Land-use regulation (LUR) was varied across SSPs in some of the models, either by changing the areas for nature conservation or areas otherwise protected from conversion to agriculture, or by modifying the ease by which non-agricultural land can be converted to agriculture. The diversity in model approaches makes it difficult to compare level of LUR across models.

Trade development (TRD) in agricultural commodities is affected in several models by (current) trade barriers such as import taxes or export subsidies and their modification toward the future. Some other models have so-called self-sufficiency ratios, which they change to reflect the storylines.

### Sensitivity analysis and design of experiment

Attributing the differences between SSP scenario results to the specific input factors is a non-trivial problem, as input factors interact in a complex system. A recently developed methodology[35,60] allows to identify the individual effects of an input factor and its summary interaction with other factors, with a limited number of model experiments, although at the expense of missing individual interactions. This sensitivity methodology was recently applied to $CO_2$ emission pathways in the SSP framework[18] and is also used here. The model inputs that vary between the various scenarios are grouped into scenario factors and the scenario results are described to be a function of a set of these input factors (here, the six driving factors POP, GDPpc, CON, PRD, LUR and TRD described above). Varying factors between a base-case (SSP2) and the alternative scenario (SSP1, SSP3 and also SSP4 and SSP5), we derive the sensitivity of results to these drivers (Supplementary Fig. 1 and Supplementary Table 2). The individual effect of one factor is derived by replacing for this factor of interest the value in the reference case (here SSP2) by the values in the alternative case (SSPX); the final effect (with opposite sign) is derived by changing all factors but the one of interest (Supplementary Fig. 1). The difference between the individual effect and the final effect is described as the interaction effect, and describes to what extent the effect of a factor is influenced by other scenario factors. This approach does not explicitly elucidate interaction between single factors, for which much more permutations of factors would be required, but reduces the number of required model runs to identify bulk interaction of one factor with the other factors. It has been shown that this computational shortcut is an adequate approximation to the full matrix of interactions, at the expense of ignoring each single interaction term[35,60]. In general, and as a plausibility test, the sum of individual effects is larger than the total difference between two scenarios

and the sum of final effects is smaller than the total difference between two scenarios.

The advantage of the sensitivity approach applied here[18] is that it requires less simulations than full-factorial analysis (GCAM, >30,000 simulations[61]), and compared with simpler approaches such as a Kaya decomposition[62] it can account for interaction between drivers and is not path-dependent as the regularly applied stepwise scenario approach[8,43].

**Quantifying the size of the drivers**. The change in most drivers compared with the SSP2 default case differs across models, as only GDP per capita (GDPpc) and population (POP) had been harmonized for the SSPs[17]. The quantification of all other storyline elements was done by the modeling groups individually. In order to quantify the size of a driver compared with SSP2, we use the most proximate result variable. For example, to estimate the size of the dietary transition and consumption preferences (CON) in SSP1 compared with SSP2, we use the change in food consumption of livestock products per capita in the scenario SSP2_CON1, i.e., the individual effect of the factor CON on the most proximate result variable, namely per capita consumption of livestock products. Likewise, we use the change in total trade volume for the driver trade development (TRD), in the scenario SSP2_TRD1, where only the TRD setting has been shifted toward SSP1. As for the sensitivity methodology, we derive both individual and final effect. All variables and scenarios to derive the size of change per driver are listed in Supplementary Table 3 and the resulting quantification is shown in Supplementary Fig. 6. To determine the sensitivities independent of driver sizes, we relate the change in output variable to the size of the driver, compared with SSP2.

## Data availability
The scenario data and sensitivity experiments used in this study are available from the corresponding author upon request.

## Code availability
Code for processing the data is available from the corresponding author upon request.

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

## Acknowledgements

S.F. and T.H. were supported by Global Environmental Research Fund 2–1702 of the Ministry of Environment of Japan. K.W. acknowledges support from the CGIAR Research Programs PIM and CCAFS. B.L.B. and J.D. were supported by the European Union's Horizon 2020 research and innovation program under grant agreement number 689150 (SIM4NEXUS). We thank Giacomo Marangoni for discussing methodological aspects with us in December 2016. Model comparison and analysis was supported and facilitated by the AgMIP global economics group.

## Author contributions

E.S., P.H. and H.L.-C. initiated the experiment and all authors contributed to its final design. W.J.v.Z., A.T., H.V., T.H., P.K., B.L.B., F.H. and D.M.-D. implemented the scenarios and provided model output data. K.C., J.C.D., S.F., P.H., H.L.-C., H.v.M., A.P. and K.W. provided feedback to the manuscript text. W.J.v.Z. prepared the figures. E.S. and H.V. wrote the paper and all authors contributed to the analysis.

## Additional information

**Competing interests:** The authors declare no competing interests.

