## [Peer Review File · Nature Communications]

Reviewers' comments:

Reviewer #1 (Remarks to the Author):

The study is presented as an exploration of how different long term drivers determine land use and per capita food availability projections (food availability should't then go in the title?). This is done under three modelled 'storylines' (scenarios) of already published Shared Socio-economic Pathways (SSPs), with focus here three of them: SSP2 used as a baseline, SSP1 for an environmental conscious future, and SSP3 as the opposite, a regional rivalry one.

The paper serves as a useful summary and comparison of six Integrated Assessment Models (IAMs— check whether the acronym is explained; could not find it), widely used by different global change workers, and as a gate to previous studies along the same lines, foremost those by Alexander Popp and colleagues. The added value of this paper is that it performs sensitivity analyses of different models to the scenarios using their 'elasticity', i.e. response per unit of change in each driver (such as GDP or population size). Moreover, these elasticities are presented as pure (individual) driver effects, interactive effects, and final effects compounding individual and interactive effects.

Presentation of the models and narratives is clear, and rightfully acknowledges the difficulties in harmonization of what necessarily are different logics under different scenarios, built by different groups and with different objectives. Supplementary materials are very helpful in this respect.

My only caveat regarding contents relates to land use regulation: LUR is found to have an overall important but variable effect which, besides of the different models' logics, is attributed to the difficulties in evaluating the proper size for this driver. However, in my opinion this deserves further discussion here (and future work elsewhere) in terms of the important institutional and cultural differences between countries regarding LUR effectiveness to slow down agricultural expansion, particularly near the tropics, and to eventually achieving land savings (e.g. Volante et al. Land Use Policy 55: 154, 2016; Sans et al. Land Use Policy 70: 313, 2018; Pellegrini & Fernández, PNAS 115: 2335, 2018; Phalan, Sustainability 2018, 10, 1760).

Reviewer #2 (Remarks to the Author):

The authors present their impressive work on key determinants of global land-use futures. They explored and compared the impacts of land change drivers on the modeling outputs of several global change models. The analysis was carried out under the SSPs scenarios, which is essential to the understanding of future global change. My specific comments on this paper are listed below:

- (1) This paper is entitled “key determinants of global land-use futures”. While the authors have explicitly stated that only socioeconomic drivers are considered, the title may be refined more specifically as “key socioeconomic determinants”. However, I still feel a little confused about the main topic of this paper. Most of the contents are sensitivity analysis of global change models with respect to the changes of input factors, and only a small part focuses on the discussion of determinants of future land change. In my opinion, if the main topic is sensitivity analysis for global models, then more discussion should be on the implications of global change modeling. If the main topic is to identify the key determinants, then I recommend adding discussion relating to land use policies according to the finding of land use change determinants. If this is the case, perhaps more analysis is required because the key determinants may vary from one region to another in the world.
- (2) This research is based on the sensitivity approach that is used in Marangoni et al. (2017). I’ve read the article of Marangoni et al. (2017) and I found the submitted manuscript has an almost identical structure of content organization. Marangoni et al. (2017) have mentioned the primary sources of the sensitivity approach, i.e. a modified order quantity (EOQ) model developed by Emanuele Borgonovo (European Journal of Operational Research, 200(1), 127-138). This reference is missing in the manuscript and I think it should be cited and clarified. If the Method section doesn’t have enough space, then more details of this model can be added in the SI file.
- (3) This study investigates the impacts of drivers with respect to individual, final and interaction effects. For a better understanding, a brief definition of these three types of effects can be added in the manuscript.
- (4) The contribution of this paper should be clearly stated. From the perspective of identifying key land use change drivers, it seems that all findings are too broad and nothing new. It is not clear how these findings can support land use policies either at global scale or regional scales.
- (5) Some minor points: Use either ‘land use’ or ‘land-use’ consistently throughout the manuscript. ‘Pasture land’ is different from ‘grassland’. I recommend using ‘Pasture land’ throughout the manuscript. Line 107, Fig. S2 => Fig. 2.

Reviewer #3 (Remarks to the Author):

I am a somewhat appropriate reviewer because I am a land change scientist who has some experience in global scenario modeling several years ago. I have not been directly involved with any of the models that the manuscript describes. If there were subtle errors in the submitted

manuscript, then I would be unlikely to catch them. Overall, the manuscript is well written, in my opinion as a native English speaker. Most of my comments below concerning how to polish the language even more, and how to make the figures more interpretable.

Line 11 in the abstract should change from “mostly arises from” to “arises mostly from” because mostly modifies from, not arises.

Line 12 in abstract should change from “different representations” to “various representations” because the word “different” can induce some readers wonder “Different than what?” which is not what the author intend. I think the authors mean various. The submitted manuscript has lots of examples where “various” would be a better word than “different”.

The manuscript is not consistent in its usage of “land use” and “land-use”. I have been told to use “land-use” when it is used as an adjective, and “land use” when it is used as a noun.

The submitted manuscript uses the word “significant” four times. A scientific paper should use the word “significant” if and only if it gives a small p-value for a conventional statistical hypothesis test in inferential statistics, where small means less than a specific alpha-level, such as 0.05 or 0.01. This manuscript should not use the word “significant” because this manuscript never performs a hypothesis test using inferential statistics. If the paper means substantial, important, or large, then the paper should use the words substantial, important, or large.

In line 33, move the word “only” to immediately before 5%, because only modifies 5%, not increased.

In line 258, move the word “only” to immediately before 2%, because only modifies 2%, not lead.

In line 58, change “but” to “and”.

It is more difficult than necessary to compare the areas of Cropland versus Pasture in figure 1, because the axes are on different scales. It would be easier to compare if the axis for Pasture matched the axis for Cropland where each additional tick mark is an increment of 100 Mha.

There is no need for the decimal place of zero and the redundant % symbols on the horizontal axes of figures 2 and 3.

Figures 2 and 3 shows three shades of bars: Individual effects (wide, light colored bars), final effects (thin, dark colored bars) and interaction effects (shaded bars). However, I am not certain where “shaded” exists with respect to light and dark. Does “shaded” mean darker than dark, or between dark & light, or lighter than light? The caption must clarify this, perhaps by changing the word “shaded” to something else. Maybe the designer can use patterns rather than shades.

Figure 3 is misleading because the distance that represents a change of 5 on the horizontal axis for SSP1 is longer than the distance that represents a change of 5 on the horizontal axis for SSP3. If a single distance represents a single increment, then it would be clearer to compare SSP1 to SSP3.

Figure 4 must indicate what the gray bars mean. Black exists in the legend but not in the figure. Maybe the black in the legend should be gray.

I hope my feedback helps.

Reviewers' comments and reply to reviewers (marked with "-> Reply: "):

Reviewer #1 (Remarks to the Author):

The study is presented as an exploration of how different long term drivers determine land use and per capita food availability projections (food availability shouldn't then go in the title?). This is done under three modelled 'storylines' (scenarios) of already published Shared Socio-economic Pathways (SSPs), with focus here three of them: SSP2 used as a baseline, SSP1 for an environmental conscious future, and SSP3 as the opposite, a regional rivalry one.

-> Reply: *Thank you for this correct summary. The reviewer asks whether food availability should not appear in the title. We prefer to only keep this short title as land use is analyzed in more detail than food availability, and – most importantly – as the SSP were rather focused on land-use than food availability, both in design and element quantification.*

The paper serves as a useful summary and comparison of six Integrated Assessment Models (IAMs—check whether the acronym is explained; could not find it), widely used by different global change workers, and as a gate to previous studies along the same lines, foremost those by Alexander Popp and colleagues. The added value of this paper is that it performs sensitivity analyses of different models to the scenarios using their 'elasticity', i.e. response per unit of change in each driver (such as GDP or population size). Moreover, these elasticities are presented as pure (individual) driver effects, interactive effects, and final effects compounding individual and interactive effects.

-> Reply: *The acronym IAM was indeed not explained. The term was hardly used in the main text, as only 5 of the six models are (components of) IAMs, and we now removed it completely from the main text. It is only used and explained now in the method section.*

Presentation of the models and narratives is clear, and rightfully acknowledges the difficulties in harmonization of what necessarily are different logics under different scenarios, built by different groups and with different objectives. Supplementary materials are very helpful in this respect.

My only caveat regarding contents relates to land use regulation: LUR is found to have an overall important but variable effect which, besides of the different models' logics, is attributed to the difficulties in evaluating the proper size for this driver. However, in my opinion this deserves further discussion here (and future work elsewhere) in terms of the important institutional and cultural differences between countries regarding LUR effectiveness to slow down agricultural expansion, particularly near the tropics, and to eventually achieving land savings (e.g. Volante et al. Land Use Policy 55: 154, 2016; Sans et al. Land Use Policy 70: 313, 2018; Pellegrini & Fernández, PNAS 115: 2335, 2018; Phalan, Sustainability 2018, 10, 1760).

-> Reply: *This is a very valid point. Land-use regulation was identified as important determinant, but the mechanism is complex, and its implementation in the global scale models is – if at all existent – still quite simplistic. We now refer in the main text to the diverging implementation of LUR in the models (“, and models vary largely in their LUR implementation (table S2). “ And while we already stated that more work on this is*

needed, we now extended the discussion of LUR, its effectiveness, and implementation in models, and we also cite the suggested references. Now the text says:

"In some cases, drivers are not or poorly represented, such as land-use regulation, although it has a large potential to restrict future agricultural land expansion⁴⁰. Currently, global models tend to only exclude certain areas from agricultural conversion, mostly focusing on official "protected areas" and their assumed expansion according to international targets, but ignore specific national and sub-national regulations on land-use zoning or deforestation rates (e.g. sugar cane zoning and forest code land reserve quota's in Brazil), and, as also shown here, land-use regulation is necessary to assure the land saving effect of agricultural intensification³⁸. Although the effectiveness of such approaches can be poor⁴¹, these regulations, as well as land-tenure aspects, are relevant to short- and medium-term conversion quantities and locations. Therefore, future model development should include more explicitly land-use regulation and its facets at various spatial scales, and also its interaction with agricultural productivity and food security."

Reviewer #2 (Remarks to the Author):

The authors present their impressive work on key determinants of global land-use futures. They explored and compared the impacts of land change drivers on the modeling outputs of several global change models. The analysis was carried out under the SSPs scenarios, which is essential to the understanding of future global change. My specific comments on this paper are listed below:

(1) This paper is entitled "key determinants of global land-use futures". While the authors have explicitly stated that only socioeconomic drivers are considered, the title may be refined more specifically as "key socioeconomic determinants". However, I still feel a little confused about the main topic of this paper. Most of the contents are sensitivity analysis of global change models with respect to the changes of input factors, and only a small part focuses on the discussion of determinants of future land change. In my opinion, if the main topic is sensitivity analysis for global models, then more discussion should be on the implications of global change modeling. If the main topic is to identify the key determinants, then I recommend adding discussion relating to land use policies according to the finding of land use change determinants. If this is the case, perhaps more analysis is required because the key determinants may vary from one region to another in the world.

-> Reply: On "key drivers" versus "key socio-economic drivers": we introduced the term "socio-economic drivers" in the paper to make explicit that we do not address possible impacts of climate change. However, several of the determinants we include do not fall clearly in the category of "socio-economic" drivers (e.g. land-use regulation, and dietary transition). We therefore now downplay the use of "socio-economic" in the main text of the paper..

The reviewer is correct that this study is a sensitivity analysis on scenario-specific input factors and assumptions in global land-use projections, and we agree that we should not overstate the topic of our paper. However, assuming that the models contain the relevant processes and drivers, we also believe that this sensitivity gives an idea about the real importance of the drivers. Beyond that, as the ambition is to cover the relevant

aspects of global land-use change, the discussion now more explicitly points to gaps and shortcomings in current modelling. .

We now state in the introduction the added values of our study (also in response to comment (4)) and distinguish there between the sensitivity aspects, and they key determinants beyond current model implementation: (“While single model studies have identified factors that determine future land use and support a transition to a sustainable land and food system, this study allows a systematic comparison across factors and models, and to identify the interaction between these factors. Furthermore the discussion highlights gaps and shortcomings in current modelling of global land-use to guide future research priorities.”)

Further we state now explicitly in the discussion which gaps and shortcomings exist in global land modelling. See above addition on LUR, and text introduced as fourth paragraph in the discussion:

“While global land-use models have the ambition to include the relevant drivers and processes to project future large-scale land-use dynamics, the identification of key determinants of future land-use needs to account for aspects currently poorly addressed by the models. Local land management regulations, as discussed above for protected areas, are poorly represented in current models, although they can play important roles at fine scale. Land-tenure, processes of intensification and technology transfer, and integration of crop cultivation and livestock with forestry and other land uses (urbanization, recreational areas, mining) are other underrepresented factors relevant for future land projections. To identify the gaps and shortcomings in global scale models, more rigorous model validation and evaluation on smaller scales are necessary.”

(2) This research is based on the sensitivity approach that is used in Marangoni et al. (2017). I've read the article of Marangoni et al. (2017) and I found the submitted manuscript has an almost identical structure of content organization. Marangoni et al. (2017) have mentioned the primary sources of the sensitivity approach, i.e. a modified order quantity (EOQ) model developed by Emanuele Borgonovo (European Journal of Operational Research, 200(1), 127-138). This reference is missing in the manuscript and I think it should be cited and clarified. If the Method section doesn't have enough space, then more details of this model can be added in the SI file.

-> Reply: Correct, we use the approach applied in Marangoni, to which we explicitly refer, and also follow their figure style and part of the structure is adopted. We thought this makes a nice parallel, and improves accessibility, as then both CO₂ emissions and land use in the SSPs are explored in a consistent way. We go beyond the Marangoni paper by introducing the sensitivity as a function of the driver size, which allows to compare model elasticities independently of assumptions on input factors. To acknowledge the work by Marangoni and colleagues, we reference their article in the text, and also mention adoption of figure styles in all our figure captions, where relevant. Note that we now also acknowledge a discussion with Marangoni we had when setting up the study. “We thank Giacomo Marangoni for discussing some methodological aspects with us in December 2016.” Other acknowledgements are not visible to the reviewers due to double-blind review.

The primary reference for the methodology is indeed Borgonovo. We had already referred to Borgonovo 2010 (“Risk Analysis” 30, 385-399) in our reference nr 46 in the methods

section. This reference had been used twice, at the beginning and towards the end of the methods section, also with the explanation that this method allows to derive interaction between drivers in a limited set of experiments but at the expense of single interactions. We now additionally include the mentioned article in "Operational Research", and also make brief reference to Borgonovo in the main paper.

(3) This study investigates the impacts of drivers with respect to individual, final and interaction effects. For a better understanding, a brief definition of these three types of effects can be added in the manuscript.

-> Reply: following this suggestion, we added a brief description in the methodological section at the end of the introduction. ("The individual effect describes how scenario results change if a single factor of SSP2 baseline is replaced by the one used in SSP1 (or SSP3), and the final effect shows how results in the full SSP1 (or SSP3) scenario change if this single factor is changed back to the SSP2 default setting; the interaction effect is computed as the difference between individual and final effect, and shows how other factors interact with this single factors, i.e. reduce the individual effect to the final effect (see Methods).")

(4) The contribution of this paper should be clearly stated. From the perspective of identifying key land use change drivers, it seems that all findings are too broad and nothing new. It is not clear how these findings can support land use policies either at global scale or regional scales.

-> Reply: As also noted above, we now state in the introduction the added value of this paper ("While single model studies have identified factors that determine future land use and support a transition to a sustainable land and food system, this study allows a systematic comparison across factors and models, allows to identify the interaction between these factors. Furthermore, the discussion highlights gaps and shortcomings in current modelling of global land-use to set future research priorities.")

And we also highlight in the discussion the added value of this study: "Beyond these earlier studies, this systematic multi-model analysis allows comparison of the relative effect across the drivers, and across the models. Furthermore, this study adds an analysis of the interaction between factors: "

(5) Some minor points: Use either 'land use' or 'land-use' consistently throughout the manuscript. 'Pasture land' is different from 'grassland'. I recommend using 'Pasture land' throughout the manuscript. Line 107, Fig. S2 => Fig. 2.

-> Reply: There is indeed in the literature some inconsistent use of "land use" or "land-use". The English spelling rule is that if two nouns are combined, there is no hyphen, but if two combined nouns are used as an adjective, i.e. to qualify a third noun, then the two are connected with a hyphen. We try to apply the correct spelling which thus requires to use "land use" without hyphen when used as a single term, and use a hyphen when it is followed by another noun, e.g. in "land-use change", "land-use regulation" (as also requested by reviewer #3). We checked the paper to follow this rule consistently. (As an indication of this rule, you will see Wikipedia entries on "land use" and one on "land-use regulation", and also see e.g. <https://www.grammarbook.com/punctuation/hyphens.asp>).

We use "pasture area" or "pasture" now instead of "grassland".

In line 107 the reference to a figure was indeed wrong. However, we want to refer to the size of drivers, which are shown in Figure S3 (Fig.S2 => Fig.S3). For clarity, we now also refer to Fig.2 at the beginning of this section on global cropland.

Reviewer #3 (Remarks to the Author):

I am a somewhat appropriate reviewer because I am a land change scientist who has some experience in global scenario modeling several years ago. I have not been directly involved with any of the models that the manuscript describes. If there were subtle errors in the submitted manuscript, then I would be unlikely to catch them. Overall, the manuscript is well written, in my opinion as a native English speaker. Most of my comments below concerning how to polish the language even more, and how to make the figures more interpretable.

Line 11 in the abstract should change from "mostly arises from" to "arises mostly from" because mostly modifies from, not arises.

-> Reply: This has now been corrected.

Line 12 in abstract should change from "different representations" to "various representations" because the word "different" can induce some readers wonder "Different than what?" which is not what the author intend. I think the authors mean various. The submitted manuscript has lots of examples where "various" would be a better word than "different".

-> Reply: Thank you for this remark. Indeed "different" is often wrongly used instead of "various". All relevant instances of the word were replaced in the manuscript by the terms various, diverse, diverging or differing.

The manuscript is not consistent in its usage of "land use" and "land-use". I have been told to use "land-use" when it is used as an adjective, and "land use" when it is used as a noun.

-> Reply: We also want to follow this rule ("Land use", but "land-use regulation"), and corrected the remaining inconsistencies. (see also above also reviewer #2 commented on that).

The submitted manuscript uses the word "significant" four times. A scientific paper should use the word "significant" if and only if it gives a small p-value for a conventional statistical hypothesis test in inferential statistics, where small means less than a specific alpha-level, such as 0.05 or 0.01. This manuscript should not use the word "significant" because this manuscript never performs a hypothesis test using inferential statistics. If the paper means substantial, important, or large, then the paper should use the words substantial, important, or large.

-> Reply: Thank you for this remark, we fully agree and replaced or removed all instances of the word "significant".

In line 33, move the word "only" to immediately before 5%, because only modifies 5%, not increased.

-> Reply: *This has now been corrected.*

In line 258, move the word "only" to immediately before 2%, because only modifies 2%, not lead.

-> Reply: *Done.*

In line 58, change "but" to "and".

-> Reply: *This has now been corrected.*

It is more difficult than necessary to compare the areas of Cropland versus Pasture in figure 1, because the axes are on different scales. It would be easier to compare if the axis for Pasture matched the axis for Cropland where each additional tick mark is an increment of 100 Mha.

-> Reply: *This has now been corrected, we now use same Y-axis scales with 100 Mha tick marks in each, and a step of 200 units in numbers.*

There is no need for the decimal place of zero and the redundant % symbols on the horizontal axes of figures 2 and 3.

-> Reply: *This has now been corrected.*

Figures 2 and 3 shows three shades of bars: Individual effects (wide, light colored bars), final effects (thin, dark colored bars) and interaction effects (shaded bars). However, I am not certain where "shaded" exists with respect to light and dark. Does "shaded" mean darker than dark, or between dark & light, or lighter than light? The caption must clarify this, perhaps by changing the word "shaded" to something else. Maybe the designer can use patterns rather than shades.

-> Reply: *We used the wrong word, "shading" should actually have been "striped". But as this was hardly visible we now use three steps in darkness, i.e. dark, light colored, very light colored.*

Figure 3 is misleading because the distance that represents a change of 5 on the horizontal axis for SSP1 is longer than the distance that represents a change of 5 on the horizontal axis for SSP3. If a single distance represents a single increment, then it would be clearer to compare SSP1 to SSP3.

-> Reply: *This has now been corrected. Horizontal axes have now identical increments and allow better comparison between SSP1 and SSP3 (in Figures 2 and 3, and in all similar figures in the supplement). We also added grey lines at the increments in all figures which makes it easier to read the size of the bars.*

Figure 4 must indicate what the gray bars mean. Black exists in the legend but not in the figure. Maybe the black in the legend should be gray.

-> Reply: *We apologize for the confusing legend, we have now corrected the legend for the grey bar, and also improved the axis title.*

Beyond the changes described above, we also made a small change in the handling of MAgPIE results for CON. MAgPIE results for CON were counter-intuitive due to also

including water available for irrigation. This is now explained in the caption of figure 2 and the SI, and the CON effect of MAgPIE was excluded from figure 4 (also explained in the SI).

I hope my feedback helps.

-> Reply: Thank you for the helpful comments!

REVIEWERS' COMMENTS:

Reviewer #1 (Remarks to the Author):

I am satisfied by the attention paid to my comments and suggestions, including those that authors decided not to accept.

Like new text and citations regarding LUR, thank you.

Reviewer #2 (Remarks to the Author):

This is my second review and I found the manuscript has been revised pretty well. All my comments have been addressed and replied adequately. I believe that the findings of uncertainty are helpful to modelers whose interest is global land use change. I would suggest acceptance of this paper.

Reviewer #3 (Remarks to the Author):

The authors wrote a detailed and helpful response to the first review. I have a few additional ideas for improvement.

The authors should consider revising the vertical axis for Cropland on figure 1 so that its range is 800, so its range matches the range of 800 the graph directly below it for Pasture. If the authors do this, then it will make it easier for the reader to compare the variation in Cropland to the variation in Pasture. When I looked at the existing figure 1, the first thing I had to do was to go through the mental gymnastics to adjust the visual effect of the fact the present range for Cropland is 600, while the range for Pasture is 800.

Figure S4 should express Population and GDP in billions, as opposed to millions, so readers will not have to count the many zeroes.

Figure S6 should express the vertical axis in units of millions of hectares, so readers will not have to decipher the cryptic scientific notation.

I hope my feedback helps.

Robert Gilmore Pontius Jr

Reviewers' comments and reply to reviewers (marked with "-> Reply:"):

Rev #3

The authors should consider revising the vertical axis for Cropland on figure 1 so that its range is 800, so its range matches the range of 800 the graph directly below it for Pasture. If the authors do this, then it will make it easier for the reader to compare the variation in Cropland to the variation in Pasture. When I looked at the existing figure 1, the first thing I had to do was to go through the mental gymnastics to adjust the visual effect of the fact the present range for Cropland is 600, while the range for Pasture is 800.

-> Reply: Done. Thank you for this suggestion.

Figure S4 should express Population and GDP in billions, as opposed to millions, so readers will not have to count the many zeroes.

-> Reply: Done. We have adjusted the figures where relevant to avoid many zero's.

Figure S6 should express the vertical axis in units of millions of hectares, so readers will not have to decipher the cryptic scientific notation.

-> Reply: Done. We have adjusted the figures where relevant to avoid many zero's.